# Association between tuberculosis and depression on negative outcomes of tuberculosis treatment: A systematic review and meta-analysis

Paulo Ruiz-Grosso[1,2], Rodrigo Cachay[1], Adriana de la Flor[3], Alvaro Schwalb[1], Cesar Ugarte-Gil [1,3,4]*

**1** Instituto de Medicina Tropical Alexander von Humboldt, Universidad Peruana Cayetano Heredia, Lima, Peru, **2** School of Public Health, Universidad Peruana Cayetano Heredia, Lima, Peru, **3** School of Medicine, Universidad Peruana Cayetano Heredia, Lima, Peru, **4** TB Centre, London School of Hygiene and Tropical Medicine, London, United Kingdom

* cesar.ugarte@upch.pe

**Data Availability Statement:** All relevant data are within the manuscript and its Supporting Information files.

## Abstract

### Background

Depression is a common comorbidity of tuberculosis (TB) and is associated with poor adherence to treatment of multiple disorders. We conducted a systematic review to synthesize the existing evidence on the relationship between depression and negative outcomes of TB treatment.

### Methods

We systematically reviewed studies that evaluated depressive symptoms (DS) directly or indirectly through psychological distress (PD) and measured negative treatment outcomes of drug-sensitive pulmonary TB, defined as death, loss to follow-up, or non-adherence. Sources included PubMed, Global Health Library, Embase, Scopus and Web of Science from inception to August 2019.

### Results

Of the 2,970 studies initially identified, eight articles were eligible for inclusion and two were used for the primary outcome meta-analysis. We found a strong association between DS and negative TB treatment outcomes (OR = 4.26; CI95%:2.33–7.79; $I^2$ = 0%). DS were also associated with loss to follow-up (OR = 8.70; CI95%:6.50–11.64; $I^2$ = 0%) and death (OR = 2.85; CI95%:1.52–5.36; $I^2$ = 0%). Non-adherence was not associated with DS and PD (OR = 1.34; CI95%:0.70–2.72; $I^2$ = 94.36) or PD alone (OR = 0.92; CI95%:0.81–1.05; $I^2$ = 0%).

**Funding:** PRG had support from FONDECYT/ CIENCIACTIVA scholarship EF033-235-2015 and from training grant D43TW007393 awarded by the Fogarty International Center of the US National Institutes of Health.

**Competing interests:** The authors have declared that no competing interests exist.

## Conclusions

DS are associated with the negative TB treatment outcomes of death and loss to follow-up. Considerable heterogeneity exists in the definition of depression and outcomes such as non-adherence across the limited number of studies on this topic.

## Introduction

The high burden of tuberculosis (TB) on morbidity and mortality around the world constitutes a significant public health concern, especially in low and middle-income countries [1–3]. The World Health Organization (WHO) estimated that 10 million people developed active TB disease in 2017 [4]. Treatment is usually given through the Direct Observation Therapy, Short-Course program (DOTS), and cure rates have been reported at 80% in 2012 and 82% in 2018 [4,5]. DOTS program refers to an approach to the delivery of antibiotic treatment in which health personnel observe patients taking their medication daily or three times per week, this approach has been found more effective when compared to self-administered therapy [6]. However, certain factors like poverty, poor access to health services, and mental illness, including depression and substance abuse, have a negative effect on treatment adherence. Decreased compliance often leads to treatment default, which not only increases the probability of medical complications such as developing multidrug-resistant TB or death but also results in greater expenditure for the public healthcare system [7].

Mental illness is also an important global health concern, with depression affecting nearly 322 million people worldwide [8]. The lifetime risk of suffering from depression among the general population is 5.1% in women and 3.6% in men [8]. Among patients with comorbidities, the prevalence of depression is even greater; for example, 26.8% of patients with hypertension and 8–18% of patients with diabetes mellitus also struggle with depression [9,10].

Both tuberculosis and depression share common risk factors, which explains the high prevalence of their comorbidity, reported to range from 10–52% [11–13]. Their interaction is complex since one disease might contribute to the development of the other. On one hand, the increase of pro-inflammatory cytokines characteristic of depression leads to decreased activation of the cellular and humoral immune systems, which contributes to the development of TB [14]. On the other hand, TB infection causes chronic inflammation, releasing pro-inflammatory cytokines that activate brain enzymes, such as indoleamine 2, 3-dioxygenase, that degrade tryptophan and thereby limit serotonin production. Anti-tuberculosis medications may also play a role in psychiatric disease: isoniazid alters serotonin uptake and high doses of ethambutol have been associated with depression [13].

Most studies on the comorbidity of TB and depression reveal a significant correlation between depression and poor adherence to TB treatment, as well as higher rates of treatment failure, development of antimicrobial resistance, and higher mortality rates [3,12,15]. The purpose of this systematic review is to synthesize all the available data on the relationship between TB and depression and to define the impact that their association has on negative TB treatment outcomes.

## Materials and methods

### Search strategy and selection criteria

We performed a systematic review and meta-analysis that followed the recommendations in the PRISMA Statement to report our findings (S1 Checklist). Ethical approval was obtained from the Institutional Ethics Committee of Universidad Peruana Cayetano Heredia (SIDISI: 103321). The study protocol was included in the PROSPERO database (CRD42018111058).

We searched the following electronic databases: PubMed, Global Health Library, Embase, Scopus and Web of Science for English language articles published any time up to August 9, 2019: grey literature was not considered. The search strategy included the terms: "Tuberculosis", "Depression", and "Depressive disorder". Detailed search strategies for PubMed are described in Table 1. Strategies for all other databases used are included in S1 Table.

Cross-sectional, case-control, cohort and clinical trial studies with information on depression and negative TB treatment outcomes were considered for review. For the primary analysis, negative outcomes included death and loss to follow-up during TB treatment; for exploratory analysis, studies that provided information on non-adherence were also included. The diagnosis of pulmonary tuberculosis, depression, and outcomes were defined as per each study. Studies that included information exclusively on individuals under the age of 16, inpatients, non-pulmonary TB, or drug-resistant TB were excluded. For the primary analysis, only studies that provided the number of depressed and non-depressed individuals with and without negative treatment outcomes were included. For the secondary analysis, all of the available studies were included.

### Outcome definitions

The primary endpoint extracted was the number of depressed and non-depressed individuals with and without treatment outcomes in order to assess the association between depression and negative outcomes (death and loss to follow-up during the TB treatment). The secondary endpoints included all of the studies that provided at least an effect size measure and confidence interval for the association between depression and negative TB treatment outcomes; each outcome was also evaluated individually. The exploratory endpoints included non-adherence as an outcome when assessing studies.

### Data extraction and quality assessment

The titles and abstracts of all identified articles were sorted by the author's last name and were split into two halves. Both halves were independently evaluated in duplicate by two investigators (The first half by RC and AD, and the second half by AS and PRG) to determine if they met the inclusion criteria. A more inclusive approach was employed during this phase in order to increase the sensitivity of article selection for the full-text analysis.

**Table 1. Terms employed for search strategy.**

| | |
|---|---|
| # 1 | ((tuberculosis[tiab] OR "Tuberculosis"[Mesh]) OR ("tuberculosis"[MeSH Terms] OR "tuberculosis"[All Fields])) |
| # 2 | (("depressive disorder"[MeSH Terms] OR ("depressive"[All Fields] AND "disorder"[All Fields]) OR "depressive disorder"[All Fields] OR "depression"[All Fields] OR "depression"[MeSH Terms]) OR (("depression"[All Fields] OR "major depressive disorder"[All Fields] OR "major depressive episode"[All Fields]) OR ("Depression"[Mesh] OR "Depressive Disorder"[Mesh]))) |
| # 3 | 1 AND 2 Humans |

The manuscripts included for full-text analysis were sorted according to the author's last name and divided into two halves. Both halves were independently evaluated in duplicate by two investigators, who then proceeded to identify which articles met the inclusion criteria for the systematic review. Due to a high volume of articles that indirectly classified depression concomitantly with anxiety using the Kessler Psychological Distress Scale (K-10), a decision to include them in the systematic review was made. Disagreements were resolved by consensus or by a fifth reviewer (CUG).

Two assessment tools were used to evaluate the quality of the papers included in our study. Both were independently evaluated by two investigators (RC and AS) and disagreements were resolved by consensus or by a third reviewer (PRG). The Newcastle-Ottawa Scale (NOS) was used for case-control and cohort studies and the National Institutes of Health's (NIH) assessment tool for cross-sectional studies and controlled intervention studies [16,17].

### Data analysis

We estimated summary ORs with 95% confidence intervals using the random effects approach with the *metaan* command for Stata16. We assessed the heterogeneity of the effects by the Cochrane Q with a null hypothesis of no heterogeneity and inconsistency $I^2$ test with scores of 0.5 to 1 being indicative of high heterogeneity.

## Results

The database search conducted on August 9, 2019, found 2144 records after the removal of duplicates. All titles and abstracts were screened and 2081 were excluded. Fig 1 depicts the study selection process. 63 articles were assessed through a full-text evaluation and only eight met the inclusion criteria for this systematic review: four measured depression directly through the use of validated scales and four measured depression indirectly through the use of the K-10 scale for psychological distress. The total number of participants from the eight included studies was 12313.

All studies that measured depression among TB patients had a longitudinal design with the exception of *Yan et al.* which was a cross-sectional study [18]. Depression was measured using both a 5-item and full version of the Center for Epidemiological Studies—Depression (CES-D) scale, the Patient Health Questionnaire (PHQ-9) scale, and by the International Statistical Classification of Diseases and Related Health Problems (ICD-10) diagnostic criteria. Three studies measured depression at baseline and at the end of the intensive phase of TB treatment, two studies measured it at the end of treatment, one measured it at monthly intervals, and one did not provide information regarding the measurement timing of depressive symptoms. Outcomes were defined as per local guidelines, which reflected the definition of WHO recommendations for TB treatment outcomes [19]. Specifically, loss to follow-up was defined as not taking medication for at least 30 days without a recommendation from the treating physician. Analysis of TB treatment outcomes revealed a higher risk of negative outcomes, lower success rate, and worse adherence to treatment among patients who presented DS. One study reported an adjusted hazard ratio of 3.46 for treatment default or death in patients with symptoms suggestive of a major depressive episode (MDE) at baseline visit [3], while another also found an increased risk for loss to follow-up and death (RR = 9.09 and 2.99, respectively) in persons with an MDE score suggestive of depression at baseline [20]. In the largest study measuring depression, the authors showed that in comparison with individuals with low depression scores, those with high scores had a greater chance of poor medication adherence (OR = 3.67) [18]. In this study non-adherence was measured using the Morisky Medication Adherence Scale (MMAS-8)[21], another study reported a statistically significant mean difference

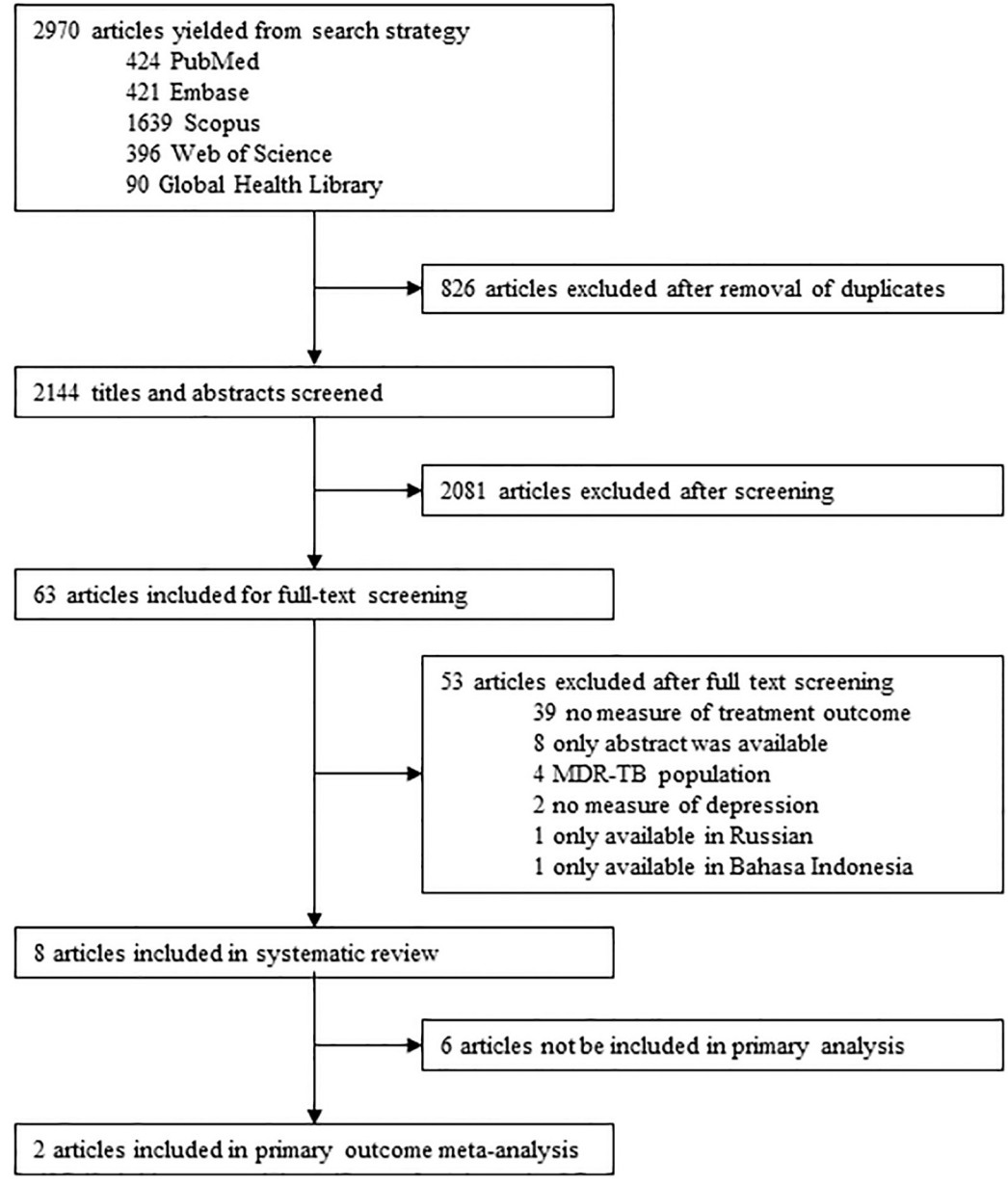

**Fig 1. Study selection.** MDR-TB: Multidrug-resistant tuberculosis.

between adherent and non-adherent individuals (7.14 vs. 2.55, p<0.001)[22]. The characteristics and results of the studies can be found in Table 2.

Four studies focused on psychological distress (PD) in patients undergoing TB treatment using the K-10 scale with different cut-off scores as detailed in Table 3. Two studies used a longitudinal design [23,24], while the other two were cross-sectional [25,26]; the latter measured PD in all TB patients that were starting treatment or re-treatment within one month of the start date. There is no widely accepted definition for non-adherence to TB treatment, and we lack strong evidence that might standardize the number of missed doses needed to impact clinical course and alter treatment outcomes. In general, adherence is measured as a proportion of missed days of treatment or using some non-TB specific scales; this might result in biased

**Table 2. Characteristics and summary of findings of articles measuring depressive symptoms.**

| Author | Year | Country | Study design | Study population | Age | Depression scale | TB diagnosis | Outcome measurements | Findings |
|---|---|---|---|---|---|---|---|---|---|
| *Govender et al.* [22] | 2009 | South Africa | Prospective cohort | 159 | Mean: 34.3 (SD: 12.2) | ICD-10 criteria | Not specified | Baseline and month 2 | Higher depression scores were associated with non-adherence to TB treatment within the intensive phase (Mean: 7.14; 95%CI 6.28–8.0 vs 2.55; 95%CI 1.88–3.22; p < 0.01). |
| *Ugarte-Gil et al.* [3] | 2013 | Peru | Prospective cohort | 325 | Median: 28 (IQR:16) in MDE group Median: 24 (IQR:9) in non-MDE group | 5-item CES-D (Cut-off score: > 6) | Sputum smear microscopy and/or culture | Baseline and monthly until treatment completion (Month 6) | MDE group presented more negative outcomes. Depression according to CES-D led to lower adherence to TB treatment. It also led to shorter survival time within the first six months of follow up (85% vs 96%) and 3.46 greater risk of loss to follow-up or death. |
| *Ambaw et al.* [20] | 2018 | Ethiopia | Prospective cohort | 648 | Mean: 30 (SD: 16) | PHQ-9 (Cut-off score: > 10) | Sputum smear microscopy and/or culture | Baseline, month 2 and 6 | Depression at baseline was associated with higher treatment loss to follow-up (3.9% vs 0.8%; p < 0.05) with significant adjusted risk ratio (aRR: 9.09; 95%CI: 6.72–12.30), as well as death (7.8% vs 1.9; p < 0.01; aRR: 2.99; 95%CI 1.54–5.78) and lower success rate (87.1% vs 96.6%; p <0.001) |
| *Yan et al.* [18] | 2018 | China | Cross-sectional | 1342 | Mean: 47.72 (SD: 17.06) | CES-D (Cut-off score: > 24) | Not specified | Not applicable | Severe depression had a greater risk of lower adherence (OR 3.67). Mild depression had greater risk of lower adherence (OR 1.92) |

CES-D: Center for Epidemiological Studies—Depression Scale. ICD-10: International Statistical Classification of Diseases and Related Health Problems (10th revision). IQR: Interquartile range. MDE: Major depressive episode. PHQ-9: Patient Health Questionnaire. SD: Standard deviation.

estimates as memory or fear of repercussions by health services might affect the accuracy of these reports. Two studies defined non-adherence as taking less than 90% of the medication in the last three to four weeks based on self-report [23,25], while another considered non-adherence as missing one DOTS appointment [23]. Outcomes of treatment, as symptom persistence, was associated with PD in the sixth month after the start of treatment (adjusted OR = 2.87) [24]. K-10 score correlated with non-adherence in one study (aOR = 1.082, CI95%:1.033–1.137)[23], however, a clear association was not found among other studies (OR = 0.92, CI95%:0.79–1.07; aOR = 0.94, IC95%:0.73–1.22)[25,26].

For the primary analyses, the only two studies (n = 973) that provided the raw number of TB patients exposed and not exposed to depression, with and without negative outcomes during TB treatment were included. We found a strong effect size (OR = 4.26, CI95%:2.33–7.79), with no evidence of heterogeneity ($I^2$ = 0%)(Fig 2).

For the three articles included in the secondary meta-analysis, summary measurements were estimated using provided effect size measures and confidence intervals. Two of the studies directly measured depression and one PD (n = 1303). We found a strong association between depression and negative TB treatment outcomes (OR = 4.05; CI95%:2.34–6.89) and no evidence of heterogeneity ($I^2$ = 0%). Exploratory meta-analysis showed that depression was associated with both death (Two articles, n = 973, OR = 2.85; CI95%:1.52–5.36) and loss to

**Table 3. Characteristics and summary of findings of articles measuring psychological distress.**

| Author | Year | Country | Study design | Study population | Age | Psychological distress scale | TB diagnosis | Outcome measurements | Findings |
|---|---|---|---|---|---|---|---|---|---|
| *Peltzer et al.* [25] | 2012 | South Africa | Cross-sectional | 4900 | Mean: 36.2 (SD: 11.5); range: 18–93 | K- 10 (Cut-off score: > 28) | Not specified | Within the first month of treatment | Non-adherence to tuberculosis medications was not associated with PD. |
| *Naidoo et al.* [26] | 2013 | South Africa | Cross-sectional | 3107 | 18–24 years: 13.5%; 25–34 years: 37.7%; 35–44 years: 28%, 45 and older: 20.8% | K-10 (Cut-off score: > 30) | Not specified | Within the first three weeks of treatment | Severe PD was associated with tuberculosis treatment non-adherence (OR:1.31 95%CI: 1.09–1.57, p < 0.01) |
| *Theron et al.* [23] | 2015 | South Africa, Zimbabwe, Zambia, Tanzania | Controlled intervention | 1502 | Median: 37 (IQR: 16) | K-10 | Culture (MGIT) | Baseline, month 2 and 6 | The median K-10 score was higher (27 vs. 21.5) among 26% of non-adherent patients. K-10 score of >30 had 2.29-fold higher risk of non-adherence to treatment. For each point increase in K-10 score, the odds of non-adherence increased by 8%. |
| *Tola et al.* [24] | 2015 | Ethiopia | Prospective cohort | 330 | Mean: 32.21 (SD: 12); range: 18–90 | K- 10 (Cut-off score: > 16) | Not specified | Baseline and month 6 | PD at the sixth month of treatment was a significant predictor of treatment outcome. |

K-10: Kessler Psychological Distress Scale. PD: Psychological Distress. TB: Tuberculosis.

follow-up (Two articles, n = 973, OR = 8.70; CI95%:6.50–11.64) and, again, no evidence of heterogeneity was found ($I^2$ = 0%). Exploratory meta-analysis showed no association between DS or PD and non-adherence (Three articles, n = 9349, OR = 1.34; CI95%:0.70–2.72) with high heterogeneity ($I^2$ = 94.36). Finally, PD alone was not associated with non-adherence to TB treatment, and this lack of association was accompanied by low heterogeneity (Two articles, n = 8007, OR = 0.925; CI95%:0.81–1.05; $I^2$ = 0) (Fig 3).

All studies underwent individual evaluation for their quality of evidence. The case-control and cohort studies were assessed by the NOS; all four studies met the Agency for Healthcare Research and Quality (AHRQ) standards for good quality evidence. The cross-sectional studies

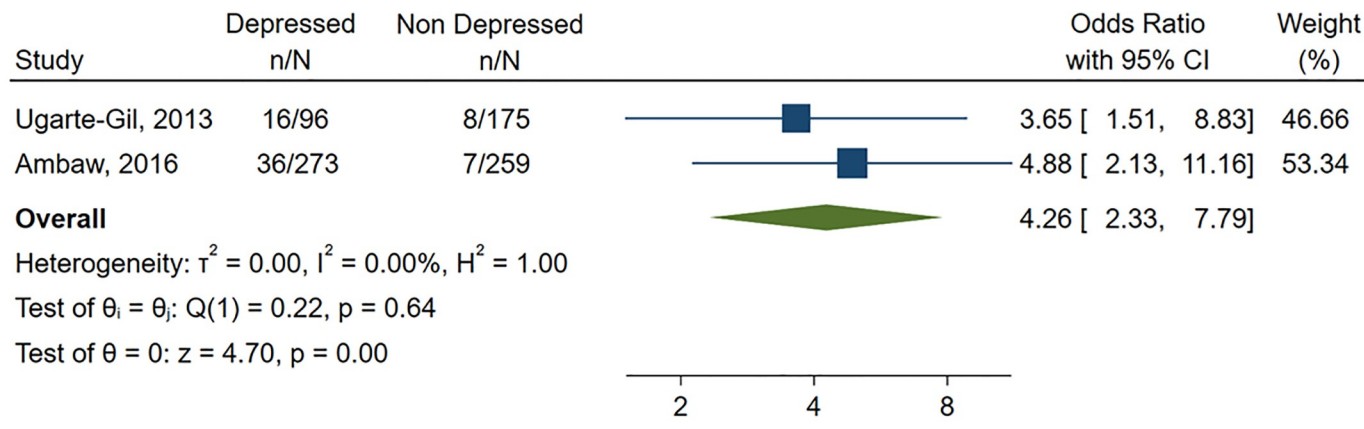

**Fig 2. Effect of depression on negative outcomes during TB treatment.**

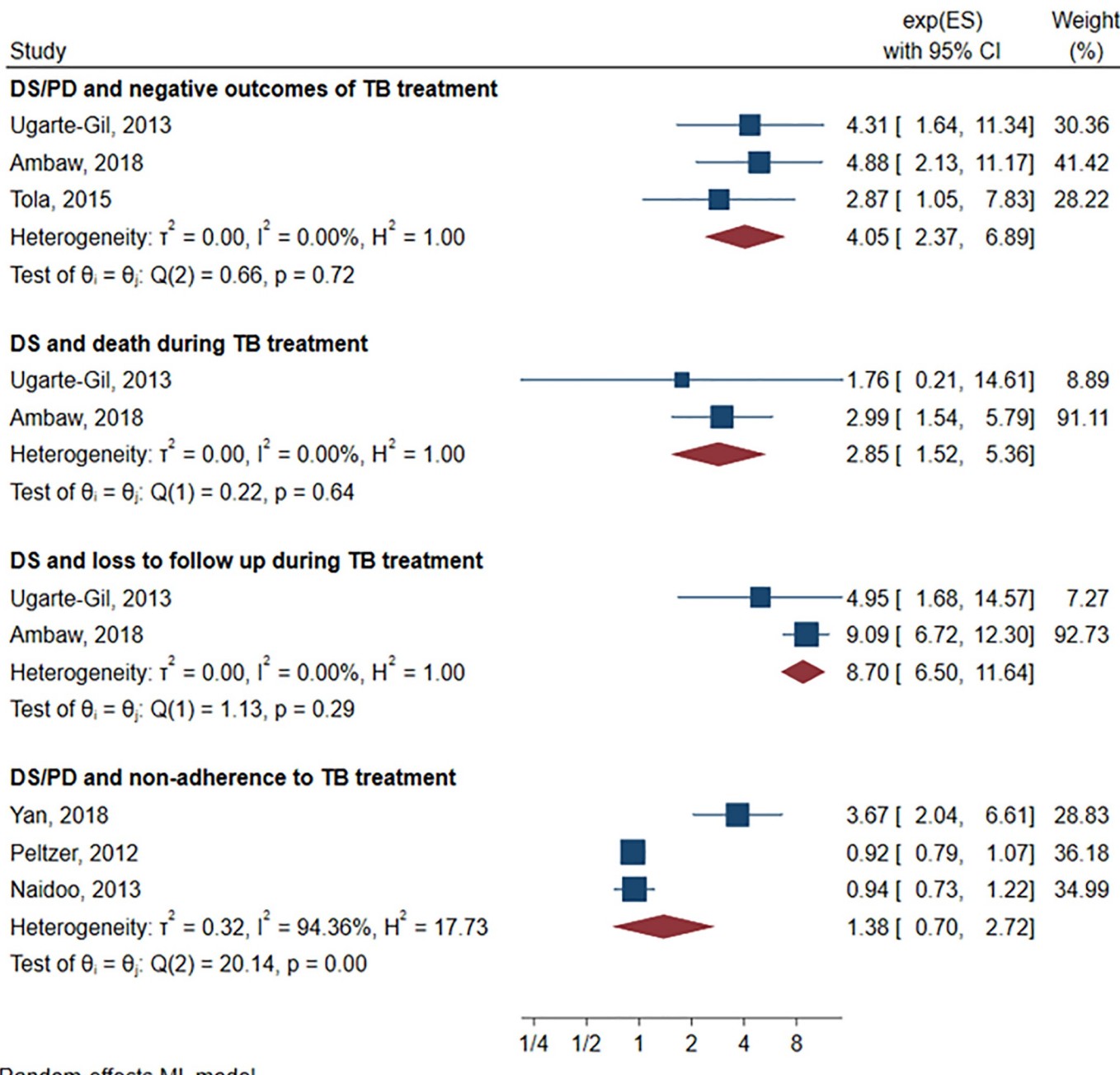

**Fig 3. Secondary and exploratory meta-analyses.** DS: Depressive symptoms. PD: Psychological distress. TB: Tuberculosis.

were also classified as good quality evidence by the NIH assessment tools. The only controlled intervention study included was rated poorly by the NIH assessment tools due to deficiencies in its description of the randomization and masking process, treatment allocation methodology, and subgroup analysis.

For the primary analysis, Cochrane GRADE scores for negative outcomes were defined as being of low quality due to the studies' observational designs. Given the overall small sample size, the quality of evidence was downgraded on the basis of imprecision, but it was not

affected by the risk of bias, inconsistency, or indirectness. Additionally, it was upgraded on account of its large effect size. Publication bias could not be established due to the low number of published manuscripts on the subject.

## Discussion

This systematic review of the impact of depression on negative TB treatment outcomes suggests that depressive symptoms are significantly associated with death and loss to follow-up during TB treatment (OR = 4.26). Across the eight studies included in the analysis, the combination of depressive symptoms (DS) and psychological distress (PD) also significantly increases the odds of negative TB treatment outcomes (OR = 4.05). Neither DS nor PD appear to be associated with non-adherence during TB treatment (OR = 1.34).

In order to properly interpret these results, several factors concerning the measurements of exposure, outcomes, possible confounders must be addressed. First, in relation to the exposure, the main analysis included studies that used both direct (DS) and indirect (PD) measurements of depression as predictors. Although depressive symptoms and psychological distress are correlated, the scales used to measure DS (one short and full version of CES-D, and PHQ-9) focused solely on depression, while the scale used to quantify PD (the K-10 scale) measured not only depressive but also anxiety-related symptoms. A high PD score thus might have actually represented mild depression with severe anxiety. This variation could explain why DS was more strongly associated with negative TB treatment outcomes than PD alone or the combination of DS and PD [27,28].

Furthermore, within each measurement of exposure, different psychometric instruments and cut-off scores were employed. In the case of studies that measured DS, there was notable variability between the scales used: a brief 5-item and the full 20-item version of the CES-D scale, and the PHQ-9 scale [3,18,20]. Others reported results using the exposure as a numeric variable alone [22,23]. Some studies used a validated cut-off score for DS compatible with MDE based on Diagnostic and Statistical Manual of Mental Disorders IV (DSM-IV) criteria [3,20], the CES-D version used by *Yan et al.* was validated with the ICD-10 criteria for mild, moderate and severe depressive disorder. Even if the validation of the instruments were applied on different populations and using a different gold standard (DSM vs. ICD-10 criteria), they were consistently validated to detect more severe forms of depression, which might explain the relative low heterogeneity found on the results. Alternatively, all of the studies on PD utilized the same K-10 scale, but each established different cut-off scores. Two studies used higher cut-off scores (>30 and >28) [25,26] when compared to *Tola et al.* (>15)[24]. Thus, when pooled together with the studies measuring DS, this lower cut-off score might have diluted the effect of the primary meta-analysis by including milder forms of DS.

Second, in terms of outcomes, the quantification of death and loss to follow-up was uniform across most of the studies reviewed, but there was heterogeneity in the definition and measurement of adherence. For death and loss to follow-up, the majority of the studies synthesized were based on DOTS clinics and used definitions similar to those found in the WHO guidelines for TB treatment [19]. Definitions of adherence were much less standard across the studies, likely due to the fact that the current WHO guidelines do not specifically define non-adherence. For instance, one study used the MMAS-8 with scores of eight or higher and lower than six suggesting high and low adherence, respectively; other studies measured adherence by asking the participants to estimate their percentage of treatment compliance in the previous three or four week interval, with less than 90% classifying as non-adherence [25,26]. While these two found no association between PD and treatment adherence, another study found a strong association between DS and adherence [18]. The lack of consistency in both the

definitions of exposure and outcome may contribute to these seemingly contradictory results. Some studies may have failed to find a significant association [25,26], however, this may be because their criteria included individuals with lower non-adherence rates when compared to *Yan et al* [18].

Third, regarding the evaluation of confounding factors, adjusted ORs were used when available for the analysis. If data on adjusted results for the analysis of negative TB treatment outcomes was not available, individual adjusted ORs for death and loss to follow-up were used instead. Most studies accounted for socio-demographic variables such as age, sex, and marital status as well as key variables related to both negative outcomes and depression, such as alcohol intake. Overall, the average age of participants in included studies ranged from 28 to 47.7 years old. On multivariate analyses, one study found older age to increase the risk of death during TB treatment [20]. Although there was no association between DS and age, *Peltzer et al.* found a higher prevalence of PD among the older population [25]. This suggests that older age might be a potential confounder on the association between DS or PD and death during TB treatment. No association was found between age and loss to follow up. However, ORs adjusted for other possible confounders such as baseline severity of depression, comorbidities, and socioeconomic status were not provided, leaving prominent unaccounted factors for negative TB treatment outcomes.

Other possible limitations of this meta-analysis include its use of cross-sectional studies and an inability to pool all adjusted ORs. A cross-sectional design is not optimal for establishing causality or providing an appropriate timeframe to study the association between exposure and outcome. These caveats aside, our findings are consistent with the literature on the role of depression on poor treatment outcomes of multiple disorders [29,30]. For example, one study found evidence suggesting that DS were related to poor adherence to HIV treatment, while another uncovered a relationship between DS and missed appointments for diabetes mellitus [31,32]. However, the causal relationship between DS and non-adherence, loss to follow-up, and death during TB treatment is likely to be complex. Depressive disorders might impair key neurocognitive functions necessary for proper adherence, such as the appreciation and reasoning components of decision making, memory, and executive function, especially when melancholic symptoms are present. This impairment, in turn, could affect awareness and the disposition to receive TB treatment. Given the evidence that such neurocognitive functions seem to improve with the remission of DS, the need to detect DS during TB treatment is all the more important [33–36].

The reviewed literature suggests that DS could be a key line of research in the efforts to improve the outcomes of TB treatment. However, in order to properly design and implement an intervention, more preliminary data must identify the most suitable course of action. For example, in order to establish the appropriate period for a timely intervention, we must first better understand how DS evolve during TB treatment and how their variation impacts treatment outcomes. This work calls for the use of universal scales with preset cut-off points to measure depression and facilitate further meta-analysis.

In addition, measuring the effect of potential mediating factors between DS and negative outcomes of TB treatment might allow us to better identify the most vulnerable populations. For example, depression and substance use disorders, which are themselves associated with both the development of TB and negative outcomes during treatment, also have a high prevalence of comorbidity [37,38]. The pathologic use of drugs such as alcohol and cocaine derivatives have been found to increase the severity of co-occurring depressive symptoms and are likely to severely disrupt cognition, further contributing to poor treatment adherence [39]. Given the multifactorial consequences of mental illness, there is no doubt that a holistic approach to patient care must be considered in order to adequately treat tuberculosis.

## Conclusions

The current evidence suggests that depressive symptoms are associated with negative outcomes during the treatment of tuberculosis. We recommend that future research focus on the longitudinal association between depression and TB outcomes, taking into account possible mediators and biological markers that might help to identify subgroups of patients that could benefit from specific interventions, including the treatment of depressive symptoms. Patient management during TB treatment should include depression screening to ensure the quality of care. We also urge that an effort be made to report results for both categorical and continuous depressive measures in order to facilitate more comprehensive meta-analyses.

## Supporting information

**S1 Checklist. PRISMA 2009 checklist.**
(DOC)

**S1 Table. Search strategy for other databases.**
(DOCX)

## Acknowledgments

We would like to thank Nicola Young for her help in reviewing this manuscript.

## Author Contributions

**Conceptualization:** Cesar Ugarte-Gil.

**Data curation:** Paulo Ruiz-Grosso, Rodrigo Cachay, Adriana de la Flor, Alvaro Schwalb.

**Formal analysis:** Paulo Ruiz-Grosso, Rodrigo Cachay, Adriana de la Flor, Alvaro Schwalb, Cesar Ugarte-Gil.

**Investigation:** Cesar Ugarte-Gil.

**Methodology:** Paulo Ruiz-Grosso, Rodrigo Cachay, Alvaro Schwalb, Cesar Ugarte-Gil.

**Project administration:** Cesar Ugarte-Gil.

**Writing – original draft:** Paulo Ruiz-Grosso, Rodrigo Cachay, Alvaro Schwalb, Cesar Ugarte-Gil.

**Writing – review & editing:** Paulo Ruiz-Grosso, Rodrigo Cachay, Adriana de la Flor, Alvaro Schwalb, Cesar Ugarte-Gil.

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
