## [Decision Letter · Decision Letter 0]

27 Nov 2019

PONE-D-19-30772

Association Between Tuberculosis and Depression on Negative Outcomes of Tuberculosis Treatment: A Systematic Review and Meta-Analysis

PLOS ONE

Dear Dr. Ugarte-Gil, 

Thank you for submitting your manuscript to PLOS ONE. After careful consideration, we feel that it has merit but does not fully meet PLOS ONE’s publication criteria as it currently stands. Therefore, we invite you to submit a revised version of the manuscript that addresses the points raised during the review process.

We would appreciate receiving your revised manuscript by Jan 11 2020 11:59PM. To enhance the reproducibility of your results, we recommend that if applicable you deposit your laboratory protocols in protocols.io, where a protocol can be assigned its own identifier (DOI) such that it can be cited independently in the future. For instructions see: http://journals.plos.org/plosone/s/submission-guidelines#loc-laboratory-protocols

We look forward to receiving your revised manuscript.

Kind regards,

HASNAIN SEYED EHTESHAM

Academic Editor

PLOS ONE

Journal Requirements:

PRG had support from FONDECYT/CIENCIACTIVA scholarship EF033-235-2015 and from the training grant D43TW007393 awarded by the Fogarty International Center of the US National Institutes of Health. The authors had full access to the study’s data and were responsible for the decision to submit for publication.

Additional Editor Comments:

Major Revision

Reviewers' comments:

Reviewer's Responses to Questions

**Comments to the Author**

1. Is the manuscript technically sound, and do the data support the conclusions?

Reviewer #1: Yes

Reviewer #2: Yes

2. Has the statistical analysis been performed appropriately and rigorously? 

Reviewer #1: I Don't Know

Reviewer #2: I Don't Know

3. Have the authors made all data underlying the findings in their manuscript fully available?

Reviewer #1: Yes

Reviewer #2: Yes

4. Is the manuscript presented in an intelligible fashion and written in standard English?

Reviewer #1: Yes

Reviewer #2: Yes

5. Review Comments to the Author

Reviewer #1: Ruiz-Grosso etal. have submitted a review and meta-analysis exploring the association of TB treatment via the DOTS program and depression resulting in negative outcomes such as non-adherence and death.

While the subject matter of the article is interesting, it occurs that the authors have been extremely careless with their citations. These must be fixed and extensively updated if this can be a published article.

On numerous occasions the cited paper appears to be incorrect and the way the papers are called out in text "Theron etal., Yan etal. Tola etal.) are papers that do not even appear in the list of references! How can the handful of papers cited for the analysis not appear in the references? The years for studies reported on appear to change between figures for example in Figure 2 Ambaw is 2016, and in Figure 3 its 2018. Even worse is that the only "Ambaw" paper in the references is a 2015 BMJ open study protocol, not even the actual findings of those authors.

Other major comments:

- The authors should expand on DOTS - what is the therapy? What drugs does it comprise? What is the length? How many times and how frequently does the patient need to see a healthcare worker? This would help identify to the reader what non-adherence means.

- Have there been any studies to show how much non-adherence is acceptable? If DOTS is a 9 month program then if drugs are taken for only 8 months, is that sufficient or still completely non-adherent?

- Page 2, line 36: State that MDR-TB is multi-drug resistant when the term is first introduced.

- Reference 7 appears to be incorrect. Double check the 121 million figure.

- Reference 8 is limited to a study in Singapore and may not apply as broadly as used in this context. The authors must find better references to support their claims.

- Reference 12 appears to be a circular reference not citing an actual research article that can be used to support the author's claims.

- Page 9 Line 198: state that 'depressive symptoms' are (DS) and psychological distress are (PD) here.

- Can the authors comment on the age of the study participants included in the studies that they used for the meta-analyses. Especially, when death is a primary outcome, it would be important to state such information.

In sum, the authors should fix the citations and the text in the figures extensively and address the additional comments outlined above.

Reviewer #2: Though the total collected dhruv's were more but those selected for the review and analysis were a small number. Understandably as the rest would not have met the required criteria.

Vital point is to address the heterogeneity in the studies including the definition of depression and treatment outcomes.

6. PLOS authors have the option to publish the peer review history of their article (what does this mean?). If published, this will include your full peer review and any attached files.

Reviewer #1: No

Reviewer #2: No

---

## [Author Response · Author response to Decision Letter 0]

12 Dec 2019

Dear Prof. Hasnain Seyed Ehtesham

Academic Editor

PLOS One

Thank you for reviewing our manuscript entitled “Association between Tuberculosis and Depression on Negative Outcomes of Tuberculosis Treatment: A Systematic Review” by Ruiz-Grosso et al. for publication as a research article in PLOS ONE. We have received your comments and have corrected the manuscript accordingly. Please find the response to the comments made bellow:

1. Please ensure that your manuscript meets PLOS ONE’s style requirements, including those for file naming.

Changes have been made to the manuscript’s style to fulfill PLOS ONE’s requirements, including headings, figure and table citations, and file naming.

2. Please include captions for your Supporting Information files at the end of the manuscript and update any in-text citations to match accordingly.

Appropriate captions and in-text citations have been made for Supporting Information. A separate file has been created for S1 Table and has been submitted.

3. Funding information should not appear in the Acknowledgments section. Please remove any funding-related text from the manuscript and let us know how you would like to update your Funding Statement section of the online submission form.

Funding-related text has been removed from the manuscript. Please update the Funding Statement section of the online submission form to read: “PRG had support from FONDECYT/CIENCIACTIVA scholarship EF033-235-2015 and from training grant D43TW007393 awarded by the Fogarty International Center of the US National Institutes of Health. The authors had full access to the study’s data and were responsible for the decision to submit for publication.”

4. Authors have been extremely careless with their citations. These must be fixed and extensively updated.

References have been added when addressing the papers included in the systematic review. Furthermore, citations have been revised to make sure they are in the correct order and contain the information that we claim in the discussion.

5. Authors should expand on DOTS. This would help identify to the reader what non-adherence means.

Two short sentences have been added in introduction which briefly describe DOTS program to help readers understand the usual outpatient setting and program employed to ensure adherence as a public health measure. 

6. Studies to show how much non-adherence is acceptable.

We have expanded the discussion on the definition and possible significance of non-adherence during TB treatment. We wish to transmit the message that there is no standardized definition for non-adherence and there is uncertainty regarding the number of missed doses needed to alter the treatment outcome. Thus, there is no level of acceptable non-adherence, however, some of the included studies do find an important association between non-adherence as defined per study. This is discussed in page 9, line 160-166, where we describe the ways non-adherence was defined in each study. Furthermore, in page 14, line 240-246, we consider this as an important limitation for the study.

7. In Page 2 Line 36: State that MDR-TB is multidrug-resistant tuberculosis when the term is first introduced.

MDR-TB is first introduced as multidrug-resistant TB.

8. Reference 7 appears to be incorrect. Double check the figure.

Reference has been updated to WHO Global Health Estimates for total number of people living with depression.

9. Authors must find a better reference than reference 8 to support their claims.

Reference has been updated to WHO Global Health Estimates for prevalence of depression among males and females.

10. Reference 12 appears to be a circular reference.

Reference has been updated to cite the actual research article that supports our claims.

11. In Page 9 Line 198: State that “depressive symptoms” is DS and “psychological distress” is PD.

“Depressive symptoms” and “psychological distress” are referenced as DS and PD in page 13, line 210-211, respectively. 

12. Comment on the age of the study participants that they used for the meta-analyses; especially for death as an outcome.

Information on age has been added to Table 2 and 3 which report characteristics and summary of findings of the included articles. This has also been addressed in page 14, line 251-262, where we specifically mention age in relation to the analysis that evaluates the association with death as an outcome.

13. Address the heterogeneity in the studies including the definition of depression and treatment outcomes.

We have reviewed the discussion regarding the heterogeneity on the definition of both the exposure (depression and psychological distress) and outcome (death, loss to follow up and non-adherence) which can be found on page 13-14, line 223-249. Depression was measured using three different scales (CES-D in full and short versions, PHQ-9), however, the scales were validated to detect severe forms of depression which might explain the low heterogeneity on the main meta-analysis results. PD was measured using the K-10 scale, albeit with different cut-off scores. Death and loss to follow up had clear definitions throughout the studies, while the variability of definitions for non-adherence has been addressed above in point 6.

We hope you find our responses to your comments to be appropriate. We look forward to hearing from you at your earliest convenience.

Yours sincerely,

César Ugarte-Gil

Corresponding Author

Instituto de Medicina Tropical Alexander von Humboldt

Universidad Peruana Cayetano Heredia

---

## [Editor Report · Decision Letter 1]

20 Dec 2019

Association between tuberculosis and depression on negative outcomes of tuberculosis treatment: A systematic review and meta-analysis

PONE-D-19-30772R1

Dear Dr. Dr. Ugarte-Gil,

We are pleased to inform you that your manuscript has been judged scientifically suitable for publication and will be formally accepted for publication once it complies with all outstanding technical requirements.

With kind regards,

HASNAIN SEYED EHTESHAM

Academic Editor

PLOS ONE

Additional Editor Comments (optional):

This manuscript is a review describing the association between TB and depression on negative outcomes on TB treatment. The Authors have carried out extensive revision addressing all comments of the reviewers. Table 2 & 3 have been modified. I recommend this manuscript for publication.
---

## [Editor Report · Acceptance letter]

26 Dec 2019

PONE-D-19-30772R1 

Association between tuberculosis and depression on negative outcomes of tuberculosis treatment: A systematic review and meta-analysis 

Dear Dr. Ugarte-Gil:

I am pleased to inform you that your manuscript has been deemed suitable for publication in PLOS ONE. Congratulations! Your manuscript is now with our production department. 

With kind regards,

on behalf of

Prof HASNAIN SEYED EHTESHAM 

Academic Editor

PLOS ONE